# Universal Representation of Generalized Convex Functions and their Gradients

## Abstract

A wide range of optimization problems can often be written in terms of generalized convex functions (GCFs). When this structure is present, it can convert certain nested bilevel objectives into single-level problems amenable to standard first-order optimization methods. We provide a new differentiable layer with a convex parameter space and show (Theorems 5.1 and 5.2) that it and its gradient are universal approximators for GCFs and their gradients. We demonstrate how this parameterization can be leveraged in practice by (i) learning optimal transport maps with general cost functions and (ii) learning optimal auctions of multiple goods. In both these cases, we show how our layer can be used to convert the existing bilevel or min-max formulations into single-level problems that can be solved efficiently with first-order methods.

## 1 Introduction

This paper targets a common need in machine learning: principled, differentiable parameterizations that encode structure beyond generic universal approximators. We study generalized convex functions (GCFs) and their gradients, and provide a practical parameterization with universal approximation guarantees and a convex parameter space.

Generic shallow and deep neural networks (SNNs and DNNs) can approximate a wide variety of functions, but they are not the best tool for all tasks. Their flexibility can come at the cost of losing important structural properties and requiring more data. In many applications we instead exploit symmetries or shape constraints. For instance, translation-invariant image classifiers are modeled with convolutional neural networks (CNNs), which are universal approximators for translation-invariant functions (Yarotsky, 2022). Similar symmetry-driven constructions have been explored in many domains (Bronstein et al., 2021).

Convex functions and their gradients are two particularly useful structured classes that have recently received increased attention from a parameterization perspective. In practice, however, many objects of interest are not convex but share key properties with convex functions. Generalized convexity relaxes convexity to capture precisely this broader structure (see Singer (1997) for a survey). When present, it can turn certain bilevel problems into single-level ones. Section 3 provides a short primer on generalized convexity, and Section 4 illustrates its role in optimal transport and mechanism design.

Despite advances in parameterizing convex functions and their gradients, relatively little work has addressed GCFs. As a result, when learning GCFs, existing methods often ignore their structural properties, especially in bilevel and min–max settings such as optimal auctions with multiple goods and optimal transport with general costs. These problems are typically much harder to solve numerically than single-level optimization, and the lack of structure-aware parameterizations has limited the scope of problems that can be tackled with theoretical guarantees.

The goal of this paper is to close this gap by developing universal approximators for GCFs and their gradients and demonstrating how this theoretical machinery can be used in practice. We show that our parameterization can recover and extend classical convex-analytic constructions while remaining amenable to gradient-based training in modern ML workflows.

**Contributions.** We summarize our main contributions:

- A differentiable parameterization of GCFs with a convex parameter space, enabling stable first-order optimization.

- Universal approximation results for both GCFs and their gradients under mild regularity conditions on the cost/surplus kernel.

- A neural-network interpretation that connects finitely $Y$-convex parameterizations to shallow architectures with `max` aggregation, suggesting deeper analogues.

- An open-source implementation with experiments on multi-item auction design and optimal transport that instantiate the theory.

The remainder of the paper is organized as follows. Section 2 reviews related work on parameterizing convex functions and their gradients. Section 3 introduces convexity and generalized convexity. Section 4 reviews how GCFs arise in optimal transport and mechanism design. While Sections 3 and 4 review existing results, Section 5 contains our novel theoretical results on the approximation of GCFs and their gradients. Section 6 interprets finitely $Y$-convex functions from a neural-network perspective, and Section 7 reports empirical results. We conclude in Section 8 with some closing remarks.

## 2 RELATED WORK

The effectiveness of neural networks is partly due to their Universal Approximation Property (UAP): any sufficiently regular function can be approximated by a large enough neural network, whether shallow or deep, a fact studied extensively in, e.g., (Hornik et al., 1989; Pinkus, 1999; Liang & Srikant, 2016; Lu et al., 2021).

Closer to our context is the literature on parameterizing and approximating convex functions. Perhaps the most natural scheme is the max-affine parameterization: any convex function can be represented as the supremum of possibly infinitely many affine functions (its subgradients). Choosing the maximum of finitely many affine functions underlies max-affine regression, as explored in (Balázs et al., 2015). (Calafiore et al., 2019) and (Kim & Kim, 2022) show how the maximum can be replaced with the Log-Sum-Exp (LSE) function to yield smooth approximations. Other works, such as (Warin, 2023) and (Amos et al., 2017), propose more sophisticated multi-layered parameterizations, while (Magnani & Boyd, 2009) study piecewise linear convex functions.

Another line of research concerns the approximation and parameterization of gradients. In contrast to the one-dimensional case, not every vector field $f : \mathbb{R}^n \to \mathbb{R}^n$ is the gradient of some scalar-valued function $g : \mathbb{R}^n \to \mathbb{R}$. When $g$ is smooth, a necessary condition for $f = \nabla g$ is the Jacobian $J_f$ being symmetric, since it equals the Hessian $H_g$, which can be hard to impose. A naive idea is to parametrize gradients by differentiating parametrizations of scalar functions (for example, by using derivatives of neural networks to approximate derivatives of functions), but this approach can fail (Saremi, 2019). Even if $f_n \to f$, it does not necessarily follow that $\nabla f_n \to \nabla f$; for example, $f_n(x) = \frac{1}{n} \sin(nx) \to 0$ while $f_n'(x) = \cos(nx) \not\to 0$. These problems make UAP results for gradients less common and much harder to obtain. As discussed in Section 5, this difficulty disappears when the functions (and their limits) are convex; (Chaudhari et al., 2024) uses this fact to construct universal approximators for gradients of convex functions. (Richter-Powell et al., 2021) and (Lorraine & Hossain, 2024) pursue a different approach by parameterizing the second derivative (the symmetric positive definite Hessian) and integrating it via neural ordinary differential equations (Chen et al., 2018a).

On the practical side, such parameterizations are routinely used to learn convex objects end-to-end: (i) convex potentials whose gradients yield transport maps or Wasserstein barycenters (Makkuva et al., 2020; Fan et al., 2020); (ii) convex value/energy functions embedded in model-based control and optimization loops (Chen et al., 2018b); (iii) convex potentials that parameterize generative flows grounded in optimal transport (Huang et al., 2020); and (iv) convex functionals over probability measures optimized directly (Alvarez-Melis et al., 2021). In these setups, "finding something convex" is the goal by design, making ICNNs and related architectures a natural fit. Complementarily, differentiable convex optimization layers embed convex programs within networks, enabling end-to-end training with convex structure (Amos & Kolter, 2017; Agrawal et al., 2019), and differentiable MPC implements convex optimization-based control policies in a learnable manner (Amos

et al., 2018). More broadly, deep declarative networks provide a unifying view of embedding optimization problems as differentiable layers (Gould et al., 2020).

In contrast to convexity, computational aspects of generalized convexity remain less explored. For surveys of the mathematical theory, see (van De Vel, 1993; Pallaschke & Rolewicz, 2013; Singer, 1997; Rubinov, 2013). GCFs are ubiquitous in many areas of applied mathematics, particularly in optimal transport, matching, and game theory. After introducing GCFs in Section 3, we showcase some of their applications in Section 4. While there has been substantial work on parameterizations of convex functions and their gradients, far less has been done for GCFs. We provide analogous results for parameterization and approximation of GCFs, and establish universal approximation guarantees for generalized convex functions and their gradients (Theorems 5.1 and 5.2).

In the absence of such parametrizations, computational approaches to problems involving GCFs have often defaulted to more generic tools, sacrificing theoretical guarantees in the process.

On the mechanism design or pricing side, some works jointly learn a mechanism together with a model of buyer behavior, leading to bilevel training procedures (Rahme et al., 2020; Nedelec et al., 2019; Shen et al., 2018). Other approaches rely on multi-agent reinforcement learning to learn mechanisms and agent policies simultaneously, which also induces a bilevel structure ((Zheng et al., 2022), (Koster et al., 2022)).

On the OT side, the computational difficulties arise from enforcing the marginal constraints: in the primal formulation, the transport plan must have prescribed marginals, while in the dual formulation the potentials must satisfy the Kantorovich inequality. To avoid handling these constraints explicitly, several works embed them into unconstrained min-max objectives. On the primal side, (Rout et al., 2021) learn a transport map through an adversarial consistency objective that implicitly enforces the pushforward constraint. On the dual side, (Makkuva et al., 2020; Korotin et al., 2022b;a) introduce min-max formulations in which the Kantorovich feasibility constraint is encoded directly in the objective through neural parameterizations of the dual potentials.

## 3 BACKGROUND

Throughout this paper we assume $X$ and $Y$ are compact subsets of Euclidean spaces. We denote arbitrary subsets of $S$ with a $\tilde{S}$. The exposition below recalls facts about convexity and then provides their generalized analogues used in the paper.

### 3.1 CONVEXITY

Let $f : X \to \overline{\mathbb{R}}$ be an extended-real-valued function. The Legendre transform (convex conjugate) of $f$ is a function on the dual space $X^*$ defined by

$$f^* : X^* \to \overline{\mathbb{R}} \qquad f^*(y) = \sup_{x \in X}\{\langle x, y \rangle - f(x)\}$$

The Legendre transform is central to convex analysis: a function is convex if and only if it coincides with the Legendre transform of another function. Equivalently, each convex function may be represented as the upper envelope of its affine supporting functions (its subgradients).

### 3.2 GENERALIZED CONVEXITY

Since the Legendre transform is defined via a supremum over affine functions, a natural generalization replaces the inner product with a more general bivariate function $\Phi : X \times Y \to \mathbb{R}$ that we call a kernel. Everything that follows depends on the choice of kernel $\Phi$ but we suppress it in the notation for brevity.

We also allow the supremum to be taken over a subset $\tilde{X} \subseteq X$. When $f$ is defined on $\tilde{X}$, we denote its $\tilde{X}$-transform by a superscript $\tilde{X}$:

$$f^{\tilde{X}} : Y \to \overline{\mathbb{R}} \qquad f^{\tilde{X}}(y) = \sup_{x \in \tilde{X}}\{\Phi(x, y) - f(x)\}$$

When $\Phi(x, y) = \langle x, y \rangle$ and $\tilde{X} = X$, these definitions recover the classical Legendre transform. Unlike standard convexity theory, we allow for functions defined on a subset of $X$. We say a function $f$ is $\tilde{X}$-convex if it is the restriction of a $\tilde{X}$-transform to the domain of $f$:

$$f = (g^{\tilde{X}})_{|\mathrm{dom}(f)}$$

Finally, denote the set of $\tilde{X}$-convex functions defined on $\tilde{Y}$ by $\mathcal{C}^{\tilde{X}}(\tilde{Y})$. Define $\tilde{Y}$-transform, $\tilde{Y}$-convexity, and $\mathcal{C}^{\tilde{Y}}(\tilde{X})$ analogously.

Appendix A.1 proves some known properties of generalized convexity analogues to those of standard convexity. Those properties are used in the proofs of theorems in later sections.

## 4 APPLICATIONS OF GENERALIZED CONVEXITY

We will show how certain optimal transport and auction design problems can be framed as finding the right generalized convex function.

### 4.1 OPTIMAL TRANSPORT

An optimal transport problem concerns relating a distribution of mass $\mu \in \Delta(X)$ on one space to a distribution of mass $\eta \in \Delta(Y)$ on another space in a cost-minimizing way.

Since minimizing a cost is equivalent to maximizing its negation, we shall work with the surplus kernel $\Phi(x, y) = -c(x, y)$ and frame the problem as maximization.

$$\sup_{\pi \in \Pi(\mu,\eta)} \mathbb{E}_\pi[\Phi(x, y)] \quad \text{(1a)} \qquad \begin{aligned} \inf_{\phi:X \to \mathbb{R},\ \psi:Y \to \mathbb{R}} & \quad \mathbb{E}_\mu[\phi(x)] + \mathbb{E}_\eta[\psi(y)] \\ \text{s.t.} \quad \forall(x, y): & \quad f(x) + g(y) \geq \Phi(x, y) \end{aligned} \quad \text{(1b)}$$

1a states the Kantorovich problem: finding a transportation plan $\pi \in \Pi(\mu, \eta)$, where a coupling $\Pi(\mu, \eta)$ is the set of all joint distributions on $X \times Y$ with marginals $\mu$ and $\eta$. Since this is linear, it also admits a dual 1b, where $f, g$ are called the Kantorovich potentials.

For a feasible $(f, g)$ pair, $(g^\Phi, g)$ is also feasible (see A.1) and increases the value of the dual. Doing it one more time yields $(g^\Phi, g^{\Phi\Phi})$, hence it is WLOG to write the dual as an optimization over GCFs:

$$\inf_{f \in \mathcal{C}^Y(X)} \mathbb{E}_\mu[f(x)] + \mathbb{E}_\eta[f^X(y)] \tag{2}$$

Another key result from the literature is that

$$\pi(x, y) > 0 \implies \nabla f(x) = \nabla_x \Phi(x, y)$$

When $\nabla_x \Phi(x, \cdot)$ is a diffeomorphism [1], we can invert it to get the optimal transportation map $T : X \to Y$:

$$T(x) = (\nabla_x \Phi(x, \cdot))^{-1}(\nabla f(x)) \tag{3}$$

This is a generalization of (Brenier, 1991), which states that for $c(x, y) = ||x - y||_2^2$ the optimal transport map is the gradient of a convex function. See (Villani et al., 2008) for a detailed discussion.

### 4.2 MECHANISM DESIGN

Let there be outcomes $X$, each priced according to a pricing function $t : X \to \mathbb{R}$. A buyer has a type $y \in Y$ unknown to the seller affecting how much they value each outcome. Buyer's utility $u(x, y) = \Phi(x, y) - t(x)$ is the value $\Phi(x, y)$ they get from the outcome $x$, given their type $y$, minus

---

[1]In the optimal transport literature, this is called the twist condition. It can also be seen as a generalization of the single crossing condition from mechanism design (Spence, 1978; Mirrlees, 1971).

their payment $t(x)$. Since we are assuming compactness of $X, Y$, we can write the buyer's choice problem as:

$$Q(t, y) = \arg\max_{x \in X} u(x, y) = \arg\max_{x \in X} \Phi(x, y) - t(x)$$

The seller, on the other hand, receives a revenue $t(x)$ minus some production cost $P(x)$ with $P : X \to \mathbb{R}$ where $x$ should be the outcome chosen by the buyer. Hence the objective of the seller is:

$$\sup_{t:X \to \mathbb{R}} E_Y(t(Q(t, y)) - P(Q(t, y)))$$
$$\text{s.t.} \quad Q(t, y) = \arg\max_{x \in X} u(x, y) - t(x)$$

In other words, the seller needs to find a price function that gives them high profit given that the buyer is making the optimal choice $Q(t, y)$. From an ML perspective, this is exactly an adversarial training or min–max setup: the seller chooses $t$ while anticipating the buyer's best response $Q(t, y)$. As mentioned in Section 1, this bilevel formulation is what many applied papers use. However, we will show that using GCFs, we can reduce this to a much simpler single-level optimization problem. To do that, we need to first define the indirect utility function, that is the utility the buyer receives from making their best choice.

$$v(y) = \max_{x \in X} u(x, y) = \max_{x \in X} \Phi(x, y) - t(x)$$

It's easy to see that $v$ is a $\Phi$-convex function. We will show how we can write the bilevel problem as a single-level problem in terms of $v$. For that, we need the revelation principle from mechanism design (Myerson, 1981). Simply put, it states that for any mechanisms with prices $t$ and choices $Q$, there is a simpler equivalent mechanism that removes the optimization problem on the buyer's end.

Given prices $t$ and choices $Q$, we can ask the buyer directly for their types and simulate their optimal choice $Q(t, y)$ for them. This simplifies the buyer's side as the buyer has no incentive to report anything but their true type. Hence, we can convert the buyer's strategic optimization into a constraint on the seller's side. These mechanisms are called direct revelation incentive compatible (DRIC) mechanisms. Direct revelation means the buyer is only asked for their type and incentive compatible (IC) means their optimal choice is to reveal its true value. The distinction may seem superficial at first but it will allow us to rewrite the problem.

As discussed, it is WLOG to work with DRIC mechanisms so we focus on them. A DRIC mechanism consists of two objects: the price $t(y)$ now depending on the type and additionally an allocation function $a : Y \to X$ deciding what outcome to assign to each reported type. We will show that given a GCF indirect utility $v$, (IC) pins down both the price $t$ and allocation $a$.

Since the mechanism is (IC), we can apply the envelope theorem (Rochet (1987)) [2] to the indirect utility $v$ to get:

$$\nabla v(y) = \nabla_y \Phi(a(y), y)$$

Again when $\nabla_y \Phi(\cdot, y)$ is invertible (the twist condition mentioned before), we have $a(y) = (\nabla_y \Phi(\cdot, y))^{-1}(\nabla v(y))$, so the allocation is pinned down by the gradient of the indirect utility $v$. To simplify the notation, define $W(v, y) = (\nabla_y \Phi(\cdot, y))^{-1}(\nabla v(y))$ so $a(y) = W(v, y)$. We can also write the payment in terms of the indirect utility from its definitions:

$$t(y) = \Phi(a(y), y) - v(y) = \Phi(W(v, y), y) - v(y)$$

---

[2] In a nutshell, it states that for $F(x) = \max_y f(x, y)$ and $y^*(x)$ denoting the argmax, then $\nabla F(x) = \nabla_x f(x, y^*(x))$ because $\nabla_y f(x, y^*(x)) = 0$.

Thus, both the price and the allocation are determined by the indirect utility function $v$, and we can write our bilevel problem as that of choosing the right GCF, the indirect utility function $v$:

$$\max_{0 \le v \in \mathcal{C}^Y(X)} \Phi(W(v,y),y) - v(y) - P(W(v,y)) \tag{4}$$

The constraint $0 \le v$ is there to make sure the buyer would want to participate in the auction. More concretely, it means $\forall y : 0 \le v(y)$, otherwise the buyer with type $y$ would not want to participate in the auction. See (Ekeland, 2010) for a more detailed discussion.

## 5 METHOD: PARAMETERIZING GENERALIZED CONVEX FUNCTIONS

Due to the symmetry between $X$ and $Y$, we focus on parameterizing $\mathcal{C}^Y(X)$, the space of $Y$-convex functions defined over $X$. We assume that the surplus $\Phi$ is locally Lipschitz. This implies that any $Y$-convex function is also Lipschitz, as it is the supremum of a family of Lipschitz functions sharing the same Lipschitz constant.

We define a function to be finitely $Y$-convex if it is the $\tilde{Y}$-transform of some function where $\tilde{Y} \subseteq Y$ is finite. Denote the space of all finitely $Y$-convex functions defined over $X$ by $\mathcal{FC}^Y(X)$:

$$\mathcal{FC}^Y(X) = \bigcup_{\tilde{Y} \subseteq Y \wedge |\tilde{Y}| < \infty} \mathcal{C}^{\tilde{Y}}(X)$$

Parameterizing $\mathcal{FC}^Y(X)$ is straightforward. Fix a finite $\tilde{Y}$. We know that $\tilde{Y}$-convex functions are the $\tilde{Y}$-transform of another function defined on $\tilde{Y}$. Thus, the space $\mathcal{C}^{\tilde{Y}}$ can be parameterized by the finite-dimensional vector space of functions $\mathbb{R}^{\tilde{Y}}$. Our first result shows that finitely $Y$-convex functions can uniformly approximate $Y$-convex functions.

**Proposition 5.1** (Uniform approximation of $Y$-convex functions)**.** *Given any $\epsilon > 0$, there is a finite $\tilde{Y} \subseteq Y$ such that for any $Y$-convex function $f \in \mathcal{C}^Y(X)$, there exists $g \in \mathcal{C}^{\tilde{Y}}(X)$ such that*

$$|f - g|_\infty < \epsilon$$

*Proof.* Deferred to Appendix (A.2.1). It would be convenient if finitely $Y$-convex functions were also $Y$-convex. This is not a given; for example, rationals are not irrationals yet they can be arbitrarily close to them. The following proposition takes care of that.

**Proposition 5.2.** *Finitely $Y$-convex functions are also $Y$-convex.*

$$\mathcal{FC}^Y(X) \subseteq \mathcal{C}^Y(X)$$

*Proof.* Deferred to Appendix (A.2.1).

Let $\overline{S}$ denote the topological closure of $S$. Combining Propositions 1 and 2 we obtain:

**Theorem 5.1** (Density of finitely $Y$-convex functions)**.** *The finitely $Y$-convex functions are dense in the space of $Y$-convex functions:*

$$\overline{\mathcal{FC}^Y(X)} = \mathcal{C}^Y(X)$$

*Hence, our parameterization of $\mathcal{FC}^Y(X)$ is a universal approximator for $\mathcal{C}^Y(X)$.*

*Proof.* Deferred to Appendix (A.2.1).

This may not be enough, as sometimes we need to approximate the gradients of $Y$-convex functions. For example, in Section 4, we saw that allocations depend on the gradient of the GCF indirect utility function.

As shown in the literature (see (Chaudhari et al., 2024)), if $f_n \to f$ and all $f_n$ and $f$ are convex, then $\nabla f_n \to \nabla f$ where the gradient exists. We extend this result to a larger class of functions, namely semiconvex functions; see (Cannarsa & Sinestrari, 2004) for a thorough introduction.

**Definition 5.1** (Semiconvexity). A function $f : X \to \overline{\mathbb{R}}$ is semiconvex if there exists a constant $K \in \mathbb{R}^+$ such that $f + \frac{K}{2}\|x\|_2^2$ is convex. A family of functions is equi-semiconvex if they are all semiconvex with the same constant $K$.

This is a much weaker condition than convexity since sufficiently smooth functions are semiconvex on a compact domain. Intuitively, semiconvexity only requires the absence of downward kinks, since any finite negative curvature can be compensated by adding a sufficiently large quadratic term.

**Proposition 5.3** (Stability of gradients under semiconvex convergence). *If $f_n \to f$ uniformly and all $f_n$ and $f$ are equi-semiconvex, then $\nabla f_n \to \nabla f$ uniformly where the gradients exist.*

*Proof.* Deferred to Appendix (A.2.1).

This result allows us to pass from uniform approximation of functions to uniform approximation of their gradients within an equi-semiconvex family, and it is the key bridge between function-level and gradient-level universal approximation in our setting.

Hence, the natural question is: Are $Y$-convex functions semiconvex?

**Proposition 5.4** (Preservation of semiconvexity under $\tilde{Y}$-transform). *If the functions $\Phi(\cdot, y)$ are equi-semiconvex, then all $\tilde{Y}$-convex functions are also semiconvex with the same constant.*

*Proof.* Deferred to Appendix (A.2.1).

A sufficient condition for this is $\Phi$ being semiconvex or twice continuously differentiable, as we are assuming compact $X, Y$. Using this, we have:

**Theorem 5.2** (Universal approximation for gradients). *If $\Phi(., y)$s are equi-semiconvex, then $\nabla \mathcal{FC}^Y(X) = \{\nabla f : f \in \mathcal{FC}^Y(X)\}$ is dense in $\nabla \mathcal{C}^Y(X) = \{\nabla f : f \in \mathcal{C}^Y(X)\}$:*

$$\overline{\nabla \mathcal{FC}^Y(X)} = \nabla \mathcal{C}^Y(X)$$

*In other words, $\nabla \mathcal{FC}^Y(X)$ are universal approximators for $\nabla \mathcal{C}^Y(X)$.*

*Proof.* Deferred to Appendix (A.2.1).

Since finitely $Y$-convex functions are defined as a finite maximum, they are not smooth. In certain applications, we may prefer to work with smoothed versions. In the standard convex setting, some recent works have focused on replacing the maximum in max-affine regression with a smooth approximation, the log-sum-exp function $\mathrm{LSE}^\tau$, and retaining the UAP. Let the $\tilde{Y}^\tau$-transform be:

$$f^{\tilde{Y}^\tau} = \mathrm{LSE}^\tau(f(y_1), \ldots, f(y_n)) = \frac{1}{\tau} \ln \left( \sum_{i=1}^n e^{\tau f(y_i)} \right)$$

Similarly, define $FC^{\tilde{Y}^\tau}(X)$ and $\nabla FC^{\tilde{Y}^\tau}(X)$ by replacing $\tilde{Y}$-transform with $\tilde{Y}^\tau$-transform.

$$f^{\tilde{Y}^\tau}(x) = \frac{1}{\tau} \ln \sum_{y \in \tilde{Y}} e^{\tau(c(x,y) - f(y))}$$

**Theorem 5.3** (Smooth Approximation). *$\bigcup_{\tau \in \mathbb{N}} FC^{Y^\tau}(X)$ uniformly approximates $\mathcal{C}^Y(X)$. If $\Phi$ is semiconvex, then $\bigcup_{\tau \in \mathbb{N}} \nabla \mathcal{FC}^{Y^\tau}(X)$ pointwise approximates $\nabla \mathcal{C}^Y(X)$ where gradients exist.*

*Proof.* Deferred to Appendix (A.2.1).

*Remark* 5.1. This means that for any desired accuracy $\varepsilon > 0$, we can uniformly approximate any $Y$-convex function on X by choosing a sufficiently large temperature $\tau$ and an appropriate finite $\tilde{Y}$. Similarly, if $\Phi$ is semiconvex, we can approximate the gradients of $Y$-convex functions pointwise (where they exist) by gradients of smoothed finitely $Y$-convex functions. In practice, this allows us to work with smooth models while retaining the approximation guarantees for both functions and gradients.

## 6 COMPARISON TO FULLY CONNECTED LAYERS

A neuron $h_i$ in a hidden layer computes an affine transformation of its input $x$ followed by a nonlinear activation function $\sigma$. These outputs are aggregated for the next layer by an aggregation function $a$, which can be a simple sum or a more complex operation such as max pooling.

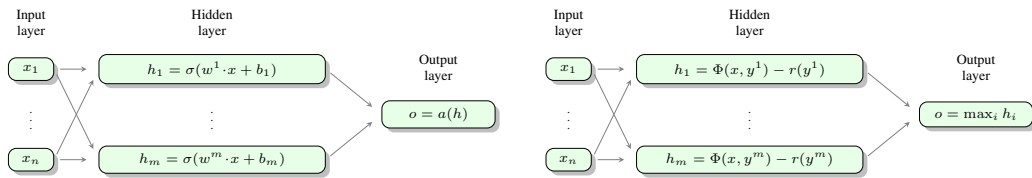

Figure 1: Comparison between finitely convex functions and neural networks: The left side shows a shallow neural network with an $n$-dimensional input layer, a single $m$-dimensional hidden layer, and a one-dimensional output layer. The right side shows a $\tilde{Y}$-convex function $r^{\tilde{Y}}$ with $\tilde{Y} = \{y^1, y^2, \ldots, y^m\}$.

As Figure 1 shows, this structure is similar to our definition of finitely convex functions. The key differences are: (1) the inner product is replaced by the surplus function $\Phi(\cdot, \cdot)$; (2) the activation function $\sigma$ is the identity function; and (3) the aggregation function $a$ is $\max$. The potential non-linearity needed is now shifted into the non-linear kernel rather than the activation function.

Therefore, the results presented in this paper can be seen from the perspective of neural networks. Although in theory a sufficiently wide shallow network can approximate any function that a deep network can, in practice deep networks often exhibit better generalization.

This suggests that there may be more powerful alternatives to our finitely convex functions, which could perform better in practice by trading network width for depth.

## 7 EXPERIMENTS

All experiments were run on a single commodity desktop (Intel i7-1365U, 12 threads, 20 GB RAM), so we restrict ourselves to moderate dimensions.

### 7.1 OPTIMAL TRANSPORT

Looking back at 2, with our approach solving the Kantorovich problem is contingent on $f^X$ being easy to compute. For example, when the cost is a metric, by Kantorovich-Rubinstein duality we have $f^X = -f$! Alternatively, if the marginals are product measures and the cost is additively separable, tensorization converts the $n$-dimensional conjugation to $n$ one-dimensional ones, almost trivial to solve (see Villani et al. (2008) for more information on both). Here, we focus on the latter case. We use a mixture of measures and consider two different costs: the quadratic cost $||x - y||_2^2$ and its negative $-||x - y||_2^2$. Though these look similar, they lead to very different results as the former prefers to transport by minimal displacement while the latter prefers to transport by maximal displacement. In particular, computing optimal transport with non-convex costs such as $-||x - y||_2^2$ is beyond the abilities of most solvers.

Figure 2 visualizes the results. It's easy to see that the marginals are well matched. Additionally, with the quadratic cost, the transport map is not moving the mass too far while with the negative quadratic cost, the transport map is moving the mass as far as possible. This is in line with what the costs are incentivizing.

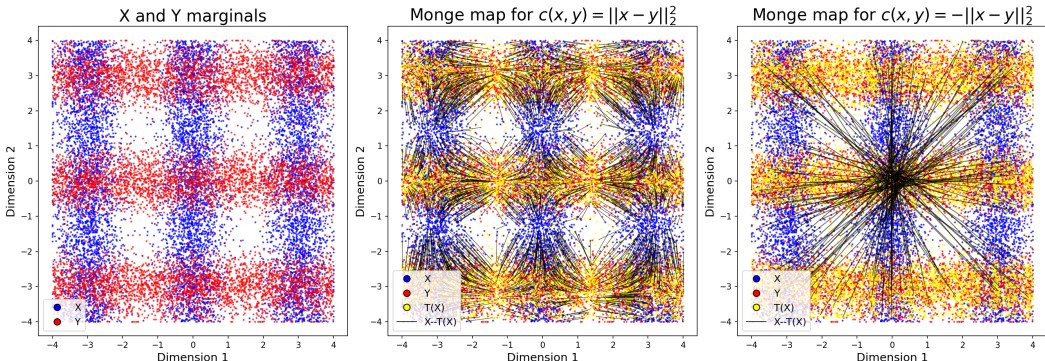

Figure 2: Visualization of the transport maps. The left plot shows the samples from the marginals. The middle and right plot show the underlying marginals and $T(X)$ where $T$ is the optimal learned transport map for $||x - y||_2^2$ and $-||x - y||_2^2$ respectively. The black lines connect $X$ and $T(X)$. As expected, the lines in the middle plot are short while lines in the right plot are as long as possible.

## 7.2 Auction Design

For auction design experiments, we will use the formulation 4 to express the seller's profit in terms of the GCF indirect utility $v$. To pin down a problem we need to specify 5 things: (1) outcome space $X$, (2) type space $Y$, (3) surplus kernel $\Phi$, (4) the production cost $P$ and lastly (5) the distribution of types. The problem with this and similar exercises is that there are very few known baselines. Settings where the optimal auction is known are limited and there are not many numerical benchmarks to compare against for more than two goods. Hence we will consider two settings: (A) a simpler setting where we have some idea about what the optimal auction should look like and (B) a more complex setting where theory can't buy us much but there are sanity checks the results should satisfy.

### 7.2.1 Setting A

We set $X = Y = [0, 1]^n$, $\Phi(x, y) = \langle x, y \rangle$, $P(x) = 0$. We can interpret this as follows. There is a single buyer and $n$ goods for sale. The types $y \in Y$ denote how much the buyer values each good. The outcomes $x \in X$ denote the probability of receiving each good. The kernel $\Phi(x, y)$ gives the expected value a buyer of type $y$ receives from outcome $x$. There are no production costs so the seller is not incurring any loss by transferring the good. Even though this setting may seem simple, it's hard to approach theoretically. Almost all the theory is concerned with the types being uniformly distributed in $Y = [0, 1]^n$ so we take the distribution of types to be uniform as well.

We know that as $n$ grows, the seller can do better by bundling some of the goods together instead of selling them separately. However, we don't know how exactly this bundling should be done and what the optimal profit is. The Straight-Jacket auction (SJa) is known to be optimal for $n \leq 6$ (Giannakopoulos & Koutsoupias, 2014) and conjectured to be optimal for $6 < n$. Even computing the revenue of this auction is complicated. (Joswig et al., 2022) provides its exact revenue for up to $n \leq 12$ so we will compare against that. Table 1 summarizes our main results. We are able to closely match SJa's revenue where available, proving the effectiveness of our approach in those cases. In addition to this, we observe two other sanity checks. The seller can always attain profit per item of $0.25$ by selling each good separately at price $0.5$. As the number of items grows, the seller should be able to do better via bundling of goods. The profit per item can never exceed $0.5$ as in that case the buyer would be better off not participating in the auction. Our findings are consistent with these expectations as the profit per good increases with the number of goods while staying within the bounds of $[0.25, 0.5]$.

Figure 3 in the appendix visualizes the learned allocations for the two-item case. Similar to SJa, the learned mechanism is neither pure bundling nor separate selling. Instead, it's a combination of both selling goods separately and together.

Table 1: mean profit per good in setting (A): The revenue of SJa is also reported when available which is the optimal for $n \leq 6$ and conjectured to be optimal for $6 < n$. Our method matches SJa's known revenue closely when available.

| Profit per Good | $n=1$ | $n=2$ | $n=4$ | $n=6$ | $n=12$ | $n=50$ | $n=100$ | $n=250$ |
|---|---|---|---|---|---|---|---|---|
| Separate Posted Pricing (analytical) | 0.250 | 0.250 | 0.250 | 0.250 | 0.250 | 0.250 | 0.250 | 0.250 |
| SJA | 0.250 | 0.274 | 0.305 | 0.324 | 0.361 | — | — | — |
| Learned Mechanism | 0.249 | 0.274 | 0.303 | 0.321 | 0.353 | 0.412 | 0.423 | 0.450 |

Table 2: mean profit per good in setting (B): There are no known optimal mechanisms or strong numerical baselines in this setting but we can see a similar trend of increasing profit per good as the number of goods increases.

| Profit per Good | $n=1$ | $n=5$ | $n=10$ | $n=20$ |
|---|---|---|---|---|
| Learned Mechanism | 1.022 | 1.202 | 1.250 | 1.253 |

### 7.2.2 SETTING B

Let $X = Y = [0, \infty)^n$ and $\Phi(x, y) = \sum_{i=1}^{n} \max(x_i - y_i, 0)$. We assume that types $y$ follow an i.i.d. log-normal distribution from a normal with mean 0 and standard deviation 0.25. Lastly, we set production cost to be $P(x) = 0.1\|x\|_2$. One interpretation is that $X$ is the quantity of each good produced and sold while $Y$ is the need of the buyer for each good. The surplus kernel $\Phi(x, y)$ captures how well the produced quantity $x$ meets the need $y$ of the buyer, giving buyer no value of the need is not met and more if it is exceeded.

Table 2 summarizes our results. Since this is a much more complicated setting, the dimensionality of the experiments is smaller compared to setting (A). As there are no known optimal mechanisms or strong numerical baselines in this setting, we focus on the qualitative properties of the learned auction and sanity checks. First, as the number of goods increases, the profit per good increases, similar to the bundling benefits observed before. Second, the profit per good is always non-negative and bounded above by the expected value of a single good under the given distribution, consistent with these expectations.

### 7.3 IMPLEMENTATION AND REPRODUCIBILITY

Both the optimal transport experiments and auction design experiments are available for reproducibility purposes as part of the `gconvex` package that implements finitely convex functions and optimizes them using PyTorch.

## 8 CONCLUSION

We developed a framework for parameterizing generalized convex functions and their gradients with a convex parameter space. Finitely $Y$-convex functions form a dense subset of all $Y$-convex functions, yielding universal approximators for both generalized convex functions and their gradients under mild regularity conditions. These parameterizations admit a neural-network interpretation via shallow architectures with max aggregation, suggesting deeper analogues.

On the applied side, our methods are implemented in the `gconvex` package and used to learn optimal transport maps with general costs and revenue-maximizing auctions with general valuation kernels, with results consistent with theory. This provides a foundation for further work in mathematical economics, optimal transport, and bilevel ML, including designing deeper finitely convex architectures and understanding when local optimization finds global optima.

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

# A    APPENDIX

## A.1    USEFUL PROPERTIES OF GENERALIZED CONVEXITY

**Theorem (Basic Properties).**    If $f, g : \tilde{X} \to \mathbb{R}$, then for all $(x, y) \in \tilde{X} \times \tilde{Y}$:

$$\Phi(x, y) \leq f^{\tilde{X}}(y) + f(x) \qquad f^{\tilde{Y}\tilde{X}}(x) \leq f(x) \qquad f \leq g \implies f^{\tilde{X}} \geq g^{\tilde{X}}$$

*Proof.* By definition of the $\tilde{X}$-transform,

$$f^{\tilde{X}}(y) = \sup_{x' \in \tilde{X}} \Phi(x', y) - f(x')$$
$$\geq \Phi(x, y) - f(x),$$

which yields the first inequality.

For the second property,

$$f^{\tilde{Y}\tilde{X}}(x) = \sup_{y \in \tilde{Y}} \Phi(x, y) - f^{\tilde{X}}(y)$$
$$\leq \sup_{y \in \tilde{Y}} f(x) = f(x),$$

where the inequality follows from the first property.

For the third property, if $f \leq g$ then

$$g^{\tilde{X}}(y) = \sup_{x \in \tilde{X}} \Phi(x, y) - g(x)$$
$$\leq \sup_{x \in \tilde{X}} \Phi(x, y) - f(x) = f^{\tilde{X}}(y).$$

$\square$

A useful analogue to the invariance under biconjugation property of convex functions is the following:

**Theorem (Generalized Biconjugation).**    Let $f : \tilde{X} \to \mathbb{R}$. Then $f$ is $\tilde{Y}$-convex if and only if

$$f = (f^{\tilde{Y}\tilde{X}})_{|\tilde{X}}$$

*Proof.* If $f$ equals its generalized biconjugate on $\tilde{X}$ then it is, by definition, the transform of $f^{\tilde{X}}$ and hence $\tilde{Y}$-convex. Conversely, suppose $f$ is $\tilde{Y}$-convex so there is $g$ with $f = (g^{\tilde{Y}})_{|\tilde{X}}$. Applying the $\tilde{X}$-transform and then the $\tilde{Y}$-transform yields

$$f^{\tilde{X}} = (g^{\tilde{Y}})^{\tilde{X}} = g^{\tilde{X}\tilde{Y}} \leq g,$$

where the last inequality is implied by the Basic Properties theorem above. Restricting back to $\tilde{X}$ and taking transforms gives

$$(f^{\tilde{Y}\tilde{X}})_{|\tilde{X}} \geq (g^{\tilde{Y}})_{|\tilde{X}} = f,$$

which together with the general inequality $f^{\tilde{Y}\tilde{X}} \leq f$ proves equality.    $\square$

*Remark* A.1. Here the notation $f^{\tilde{Y}\tilde{X}}$ means that we first take the $\tilde{Y}$-transform and then the $\tilde{X}$-transform of $f$. Thus the identity $f = (f^{\tilde{Y}\tilde{X}})_{|\tilde{X}}$ is the natural analogue of $f = (f^{**})_{|X}$ in the classical convex-conjugate setting, and the order of transforms matches the usual biconjugation convention.

As a simple corollary, distinct $\tilde{Y}$-convex functions have distinct $\tilde{X}$-transforms.

### A.2 Proofs

#### A.2.1 Method (Section 5)

**Proof of Proposition (Uniform approximation of $Y$-convex functions).** Since $\Phi$ has a Lipschitz constant $\lambda$, for any $y_1, y_2 \in Y$,

$$|y_1 - y_2| < \frac{\epsilon}{2\lambda} \implies |\Phi(x, y_1) - \Phi(x, y_2)| < \frac{\epsilon}{2}.$$

Since $f$ is $Y$-convex,

$$f(x) = f^{YX}(x) = \sup_{y \in Y}\{\Phi(x, y) - f^X(y)\},$$

where $f^X$ is $X$-convex and also has Lipschitz constant $\lambda$. Thus,

$$|y_1 - y_2| < \frac{\epsilon}{2\lambda} \implies |f^X(y_1) - f^X(y_2)| < \frac{\epsilon}{2}.$$

Since $X \times Y$ is compact and metric, it is totally bounded. So we can cover $Y$ with finitely many balls of radius $\frac{\epsilon}{2\lambda}$; let the centers be $\tilde{Y} = \{y_1, \ldots, y_k\}$. Since $\tilde{Y} \subseteq Y$, we have $f^{\tilde{Y}X} \leq f^{YX} = f$. To show the reverse inequality, observe that any $y \in Y$ is within distance $\frac{\epsilon}{2\lambda}$ of some $y_i \in \tilde{Y}$, so

$$\Phi(x, y) - f^X(y) \leq \Phi(x, y_i) - f^X(y_i) + \epsilon.$$

Therefore,

$$\begin{aligned}
f^{YX}(x) &= \sup_{y \in Y}\{\Phi(x, y) - f^X(y)\} \\
&\leq \sup_{y_i \in \tilde{Y}}\{\Phi(x, y_i) - f^X(y_i) + \epsilon\} \\
&= f^{\tilde{Y}X}(x) + \epsilon.
\end{aligned}$$

$\square$

**Proof of Proposition (Finitely $Y$-convex functions are $Y$-convex).** Let $f \in \mathcal{C}^{\tilde{Y}}(X)$ for some finite $\tilde{Y} \subseteq Y$. By definition there is a function $g_1$ on $\tilde{Y}$ such that

$$f = (g_1^{\tilde{Y}})_{|X}.$$

Define a function $g_2$ on $Y$ by

$$g_2(y) = \begin{cases} g_1(y), & y \in \tilde{Y}, \\ +\infty, & y \in Y \setminus \tilde{Y}. \end{cases}$$

Then $g_2^Y$ coincides with $g_1^{\tilde{Y}}$ on $X$, so $f = (g_2^Y)_{|X}$ and hence $f \in \mathcal{C}^Y(X)$. Since $f$ was an arbitrary finitely $Y$-convex function, this shows $\mathcal{FC}^Y(X) \subseteq \mathcal{C}^Y(X)$. $\square$

**Proof of Proposition (Stability of gradients under semiconvex convergence).** We can rely on the results concerning convex functions. Define $h_n(x) = f_n(x) + K|x|^2$ and $h(x) = f(x) + K|x|^2$. Then $h_n$ and $h$ are convex, so $\nabla h_n \to \nabla h$ uniformly where defined. Since $\nabla f_n = \nabla h_n - K\nabla|x|^2$ and $\nabla f = \nabla h - K\nabla|x|^2$, we obtain $\nabla f_n \to \nabla f$ uniformly where the gradients exist. $\square$

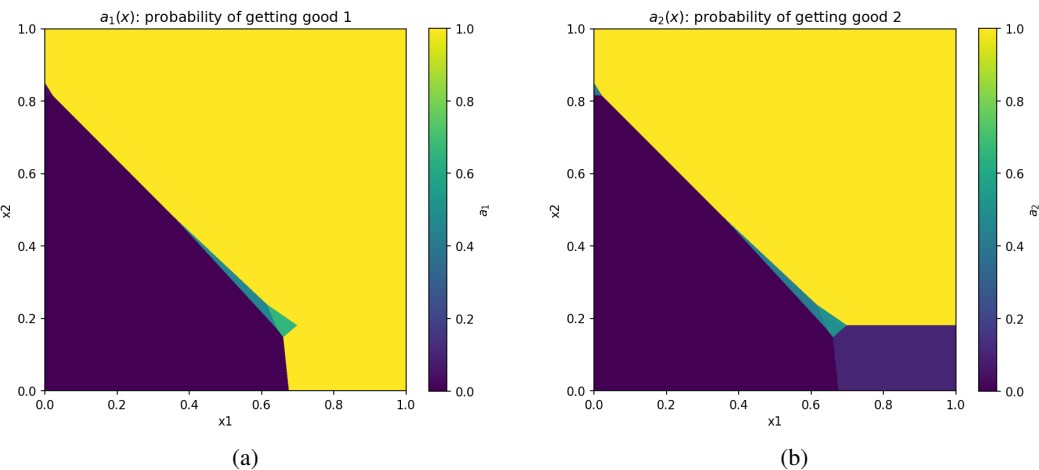

(a)                                                    (b)

Figure 3: Auction learned for the two-item case for setting (A). The allocation is neither pure bundling nor separate selling. Similar to SJa, it prices combinations of items and exhibits bunching.

**Proof of Proposition (Semiconvexity is preserved).**    If $\Phi$ has semiconvexity constant $K$, then for any $f \in \mathcal{C}^{\tilde{Y}}(X)$,

$$ f(x) + \frac{K}{2}|x|^2 = \sup_{y \in \tilde{Y}} \Phi(x, y) + \frac{K}{2}|x|^2 - g(y) $$

which is the supremum of convex functions and hence convex.                                          □

**Proof of Theorem (Density of finitely $Y$-convex functions).**    By Proposition 5.1, for any $\epsilon > 0$ and any $f \in \mathcal{C}^Y(X)$ there is a finite $\tilde{Y}$ and $g \in \mathcal{C}^{\tilde{Y}}(X)$ with $\|f - g\|_\infty < \epsilon$. Proposition 5.2 shows $\mathcal{C}^{\tilde{Y}}(X) \subseteq \mathcal{C}^Y(X)$. Hence $\overline{\mathcal{FC}^Y(X)} = \mathcal{C}^Y(X)$.                                          □

**Proof of Theorem (Universal approximation for gradients).**    Let $(g_n)_n \subset \mathcal{FC}^Y(X)$ uniformly approximate $f \in \mathcal{C}^Y(X)$ (Theorem above). By Proposition 5.4, each $g_n$ and $f$ are semiconvex with the same constant $K$ when $\Phi$ is semiconvex. By Proposition 5.3, $\nabla g_n \to \nabla f$ uniformly where gradients exist. Thus $\overline{\nabla \mathcal{FC}^Y(X)} = \nabla \mathcal{C}^Y(X)$.                                          □

**Proof of Theorem (Smooth approximation).**    Recall that for any real numbers $a_1, \ldots, a_m$ and $\tau > 0$,

$$ \max_i a_i \leq \text{LSE}^\tau(a_1, \ldots, a_m) \leq \max_i a_i + \frac{\log m}{\tau}. $$

Fix finite $\tilde{Y}$ and define $h(x) = \max_{y \in \tilde{Y}}\{\Phi(x, y) - g(y)\}$ and its smoothed version $h_\tau(x) = \text{LSE}^\tau(\{\Phi(x, y) - g(y)\}_{y \in \tilde{Y}})$. Then $\|h_\tau - h\|_\infty \leq \frac{\log |\tilde{Y}|}{\tau}$. Combining with the uniform approximation of $\mathcal{C}^Y(X)$ by finitely $Y$-convex functions yields that $\bigcup_\tau \mathcal{FC}^{Y^\tau}(X)$ uniformly approximates $\mathcal{C}^Y(X)$. Moreover, when $\Phi$ is semiconvex, each $h_\tau$ is smooth and semiconvex; as $\tau \to \infty$, $h_\tau \to h$ uniformly and $\nabla h_\tau \to \nabla h$ pointwise wherever the maximizer is unique (a.e. under mild conditions), giving pointwise density of gradients.                                          □

