# OpenReview forum: "UNIVERSAL REPRESENTATION OF GENERALIZED CONVEX FUNCTIONS AND THEIR GRADIENTS"
_ICLR.cc/2026/Conference — Submitted to ICLR 2026_

### Official Review · Reviewer_PtpT · 2025-10-28

**Soundness:** 3
**Presentation:** 2
**Contribution:** 2
**Rating:** 4
**Confidence:** 4

**Summary:**

The paper introduces a new differentiable parameterization for generalized convex functions (GCFs) and their gradients, extending prior work on convex and input-convex neural networks. It proves universal approximation results for both functions and gradients under mild semiconvexity conditions and establishes a convex parameter space that enables stable first-order optimization. The authors connect their construction to neural network architectures, interpreting finitely convex functions as shallow networks with max aggregation.
They validate the approach on a mechanism design problem, showing it recovers known optimal auction outcomes. Overall, the work offers a unified theoretical framework linking convex analysis and learnable, structured function classes.

**Strengths:**

The main strength lies in its theoretical originality—generalizing convex-function parameterizations to the broader and practically relevant class of GCFs. The paper provides rigorous proofs and smooth differentiable variants, filling a gap in the literature on universal approximators for generalized convex functions and their gradients.
Conceptually, it bridges convex analysis with neural network design, suggesting new structured architectures that preserve generalized convexity.

**Weaknesses:**

**Novelty**: This seems to be a direct extension of the work Balazs et al 2015 for convex functions. Here, the scalar product is replaced by the function $phi(x,y)$, but the results do not present any particular challenge to prove. Besides, the experimental validation choses phi to be a scalar product, so the motivation for general $phi$ in practical problems is weak.

**Narrow empirical validation** Experiments focus solely on mechanism design and fail to demonstrate the method’s potential on broader or more recognizable machine learning tasks. The paper does not showcase improvements in adversarial training, optimal transport, or robust optimization—domains explicitly cited as motivations. As a result, the contribution may seem more mathematical than impactful for mainstream ML practice. The computational scalability and practical guidance for applying the method in high-dimensional settings remain underexplored. To strengthen the work, additional experiments on modern ML problems would be essential to illustrate the method’s real-world relevance and power.


**Clarity**: I found parts of the paper are unclear and confusing. In particular, in the section 4.2 on mechanism design and the experimental details in section 7 related to it. Specifically, I did not understand how the expected revenu of the seller is defined, why is the problem a bilevel problem, what is learned and how. A full description of the objectives to optimize, the algorithms should be provided. Although the code is provided, I could not understand it without knowing beforehand these elements.
Also, some notions are introduced but without proper reference to the literature: in particular DRIC seems to be a standard concept in economics, but no reference is provided for it. I found section 4.2 on mechanism design a bit obscure. For instance, I don't understand how the seller maximizes their revenue by optimizing the price. What is the objective function that they optimize



**Soundness**: The proof of proposition 5.1 assumes implicitly that X,Y are compact, this was stated nowhere in the main paper.

**Questions:**

- In theorem 3.2, shouldn't it be f = (f^{XY})_X instead of f = (f^{YX})_X.
- The paper focuses on approximation of the gradient  of finite approximations to these generalized convex functions, but often, it is the parameters of the approximation that we'd like to learn using gradient methods, which is different from approximating the gradient of a generalized convex function wrt to its input. What can be said about learning the parameters  of these approximations using gradient methods?

---

> ### Author Response · Authors · 2025-12-01
>
> Response to Reviewer 3 (PtpT)
> Submission: 7643
>
> Dear Reviewer,
>
> Thank you very much for your thorough and insightful review. I believe your comments on novelty, empirical scope, and clarity in the mechanism design section have been extremely helpful in improving the manuscript. I also sincerely apologize for the delayed response; as the sole author, implementing the requested changes, including new experiments, took substantial time.
>
> Below I respond to your points one by one.
>
> ------------------------------------------------------------
> Point 1
> ------------------------------------------------------------
> > Novelty: This seems to be a direct extension of the work Balazs et al 2015 for convex functions. Here, the scalar product is replaced by the function Φ, but the results do not present any particular challenge to prove.
>
> Response:
>
> The connection to Balazs et al. (2015) and related max-affine approaches is very natural and explicitly acknowledged in the manuscript. As for the novelty and theoretical challenges, I agree with you when it comes to Theorem 5.1 (universal approximation of GCFs by finitely Y-convex functions): there the proof strategy is close in spirit to max-affine arguments, taking advantage of compactness of the domain and Lipschitz continuity.
>
> However, I believe Theorem 5.2 (universal approximation of gradients) is considerably more subtle and not obvious a priori. The reason it may now look simple is that the final presentation is quite streamlined. Originally, I proved a stronger intermediate result formulated in terms of directional Lipschitz properties of the gradients; only after working through that and simplifying did it become clear how to present the core idea in its current form. A similar comment applies to Theorem 5.3.
>
> In the revision I have also slightly strengthened Theorems 5.2 and 5.3 by only assuming equi-semiconvexity of the branches $\Phi(\cdot,y)$. I believe not only that this is sufficient for convergence of gradients, but also that it is essentially necessary when the kernel is non-trivial (for each $x,y$ there exists $y'$ such that $\Phi(x,y)-\Phi(x,y')$ is not constant in $x$). Unfortunately I do not have time in this revision to provide a formal proof of this necessity claim, but I hope this perspective helps convey why I view the gradient results as non-trivial.
>
> All in all, difficulty is subjective and something hard for one reader can be easy for another. If that is the case here, I hope the new applications added to the paper help convince you of its merits.
>
> ------------------------------------------------------------
> Point 2
> ------------------------------------------------------------
> > Besides, the experimental validation choses φ to be a scalar product, so the motivation for general Φ in practical problems is weak.
>
> Response:
>
> You are absolutely right and I agree.
>
> To address this, I have expanded the experiments in two ways:
>
> 1. Optimal Transport:
>    - I now include experiments where the cost is both the classical quadratic cost ∥x−y∥² and a non-convex negative quadratic cost −∥x−y∥².
>    - The negative quadratic cost lies outside the classical convex OT framework and illustrates a setting where generalized convexity is natural and useful.
>
> 2. Mechanism Design:
>     - I have extended the original linear kernel experiments to higher dimensions (up to 250 items) to demonstrate scalability.
>    - In addition to the linear kernel Φ(x,y)=⟨x,y⟩, I introduce a second setting with a nonlinear kernel combined with log-normal types and a nontrivial production cost.
>
> These additions show that the method is not restricted to linear kernels, and that generalized Φ can be practically relevant in both OT and mechanism design contexts.

---

> > ### Author Response · Authors · 2025-12-01
> >
> > ------------------------------------------------------------
> > Point 3
> > ------------------------------------------------------------
> > > Narrow empirical validation Experiments focus solely on mechanism design and fail to demonstrate the method’s potential on broader or more recognizable machine learning tasks. The paper does not showcase improvements in adversarial training, optimal transport, or robust optimization—domains explicitly cited as motivations. As a result, the contribution may seem more mathematical than impactful for mainstream ML practice. The computational scalability and practical guidance for applying the method in high-dimensional settings remain underexplored.
> >
> > Response:
> >
> > I appreciate this concern. As mentioned above, new applications have been added to demonstrate a broader applicability of the methods. In addition, the dimensionality of the auction design problem has been increased to 250 items.
> >
> > The mechanism design primer (Section 4.2) has also been rewritten to make the connection to adversarial training more explicit by writing the seller's problem in an explicit adversarial (bilevel) form.
> >
> > That said, I agree that a full-scale benchmarking study across adversarial training, robust optimization, and large standard ML datasets would be valuable but is beyond the scope of this revision. I now explicitly note in the Conclusion that exploring these broader applications and high-dimensional architectures (e.g., convolutional Φ, deep finitely convex networks) is an important direction for future work.
> >
> > ------------------------------------------------------------
> > Point 4
> > ------------------------------------------------------------
> > > Clarity: I found parts of the paper are unclear and confusing. In particular, in the section 4.2 on mechanism design and the experimental details in section 7 related to it. Specifically, I did not understand how the expected revenu of the seller is defined, why is the problem a bilevel problem, what is learned and how. A full description of the objectives to optimize, the algorithms should be provided.
> >
> > Response:
> >
> > Upon re-reading Section 4.2 I fully agree with your assessment.
> > In the revised manuscript:
> >
> > - In the Applications section (mechanism design subsection), I now start with a clear bilevel formulation:
> >   - The buyer chooses x to maximize u(x,y)=Φ(x,y)−t(x) for their type y.
> >   - The seller chooses a pricing function t to maximize expected revenue E[t(Q(t,y))−P(Q(t,y))], explicitly showing that the buyer’s optimal response Q(t,y) appears as a constraint.
> > - I then explain how the revelation principle and direct revelation incentive compatible (DRIC) mechanisms convert this bilevel problem into a single-level problem over an indirect utility function $v(y)$, which is generalized convex.
> > - I show explicitly how allocations and payments are recovered from $v(y)$ using generalized convexity and the envelope theorem, and I write the seller’s objective as a single-level optimization problem over $v$.
> > - I cite additional references for the envelope theorem and the revelation principle. An additional footnote also briefly explains the role of the envelope theorem and why it is useful here.
> > - In the Experiments section, I clearly state how the problem maps to the theoretical framework of Section 4.2.
> >
> > These changes are intended to make the bilevel nature, its reduction via GCFs, and the learning procedure much more transparent.
> >
> > ------------------------------------------------------------
> > Point 5
> > ------------------------------------------------------------
> > > in particular DRIC seems to be a standard concept in economics, but no reference is provided for it.
> >
> > Response:
> >
> > You are correct. In the revised version, I now explicitly cite standard mechanism design references when introducing DRIC mechanisms and the revelation principle (e.g., Myerson’s classical work and Ekeland’s notes on optimal transportation and mechanism design). Additionally, I've rephrased my explanation of DRIC for more intuition and clarity.

---

> > > ### Author Response · Authors · 2025-12-01
> > >
> > > ------------------------------------------------------------
> > > Point 6
> > > ------------------------------------------------------------
> > > > I found section 4.2 on mechanism design a bit obscure. For instance, I don't understand how the seller maximizes their revenue by optimizing the price. What is the objective function that they optimize.
> > >
> > > Response:
> > >
> > > This is closely related to Point 4. The obscurity stemmed from not writing the seller’s problem in full detail and not clearly connecting it to the indirect utility v.
> > >
> > > In the revised section, I explicitly:
> > >
> > > - Write the seller’s profit as the expected payment minus production cost, with the buyer’s choice explicitly enforced.
> > > - Show the bilevel structure and the DRIC reduction.
> > > - Derive a single-level objective in which the seller chooses a GCF v(y) (subject to individual rationality constraints) and revenue is expressed in terms of v and its gradient via the envelope theorem and the generalized conjugacy structure.
> > >
> > > This makes the seller’s objective and the role of v explicit in the text.
> > >
> > > ------------------------------------------------------------
> > > Point 7
> > > ------------------------------------------------------------
> > > > Soundness: The proof of proposition 5.1 assumes implicitly that X,Y are compact, this was stated nowhere in the main paper.
> > >
> > > Response:
> > >
> > > Thank you for catching this. In the revised manuscript:
> > >
> > > - I explicitly assume at the beginning of the Background section that X and Y are compact subsets of Euclidean spaces.
> > >
> > > ------------------------------------------------------------
> > > Point 8
> > > ------------------------------------------------------------
> > > > In theorem 3.2, shouldn't it be f = (f^{XY})_X instead of f = (f^{YX})_X.
> > >
> > > Response:
> > >
> > > The theorem in question is the generalized biconjugation theorem(now moved to the appendix). My notation uses superscripts to indicate which set is used in the supremum, so f^{Y X} means “take the Y-transform and then the X-transform”.
> > >
> > > The intended statement is:
> > >
> > > - f is Y-convex if and only if f=(f^{Y X}) restricted to X.
> > >
> > > This is consistent with your suggested order once one translates between the notations. In the revision, I clarify the notation in the statement and proof of this theorem in the appendix, so the order of transforms is less ambiguous.
> > >
> > > ------------------------------------------------------------
> > > Point 9
> > > ------------------------------------------------------------
> > > > The paper focuses on approximation of the gradient of finite approximations to these generalized convex functions, but often, it is the parameters of the approximation that we'd like to learn using gradient methods, which is different from approximating the gradient of a generalized convex function wrt to its input. What can be said about learning the parameters of these approximations using gradient methods?
> > >
> > > Response:
> > >
> > > If I understand your question correctly, you are asking about the connection between the gradient approximation results and learning the parameters of the finite approximations using gradient-based methods.
> > > - We have shown that gradients of finitely convex functions can approximate gradients of GCFs.
> > > - This means that using first-order methods we can learn the parameters of finitely convex functions (the finite support $\tilde{Y}$ and the intercepts) to approximate the gradient of any GCF.
> > > - For example, in the auction design setting, the profit of the seller depends on both the indirect utility $v$ and its gradient $\nabla v$. Using our method, we can find parameters that approximately maximize the profit over this structured class.
> > >
> > > While I do not prove new optimization-theoretic results about parameter learning, the framework has a favorable property:
> > >
> > > - The parameter space (values on a finite $\tilde{Y}$ and intercepts) is convex, so first-order methods are applicable.
> > >
> > > Unfortunately I am not able to theoretically guarantee convergence to global optima, but the experiments show that in practice the method works well.
> > >
> > > Once again, thank you for your very careful and constructive review. Your comments have led to substantive improvements in the clarity, rigor, and empirical scope of the paper.

---

### Official Review · Reviewer_MUph · 2025-10-29

**Soundness:** 4
**Presentation:** 2
**Contribution:** 2
**Rating:** 4
**Confidence:** 3

**Summary:**

This paper attempts to establish a unified theory for Generalized Convex Functions (GCFs) and their gradients in the context of parameterization and universal approximation. It proposes a differentiable finite representation — finitely Y-convex functions — as an alternative to neural network architectures. The authors claim that:
- They provide universal approximation theorems for GCFs and their gradients;
- They prove gradient convergence under semiconvexity assumptions;
- They demonstrate an application to multi-item mechanism design.

**Strengths:**

- It tackles an important theoretical gap by attempting to unify convex and generalized convex structures within a learnable, differentiable framew.
- The link between generalized convexity and bilevel optimization (e.g., mechanism design, optimal transport) is conceptually appealing.
- The mathematical framework, if rigorously developed, could provide a new lens for understanding structure-preserving function approximation.

**Weaknesses:**

- In the introduction, the paper argues that GCFs can transform bilevel problems into single-level optimization problems, yet all subsequent examples (e.g., mechanism design and optimal transport) already presuppose the existence of GCF representations via the Φ-transform.
This creates a circular argument:
  “We study GCFs because they simplify bilevel optimization; we know they simplify it because we assume the problem already admits a GCF form.”

- The notion of generalized convexity was formalized long ago in Singer (1997) and Rubinov (2013). The paper fails to clearly articulate whether its novelty lies in the parameterization of the function space or in an extension of existing approximation theorems. This ambiguity undermines the conceptual contribution.

- Theorem 5.1 (GCF UAP) merely asserts that “Proposition 5.1 + 5.2 ⇒ density,” yet Proposition 5.1’s proof relies on Φ being globally Lipschitz.
Since Φ is only assumed to be locally Lipschitz, there is no guarantee of a global constant on compact domains.
→ Therefore, the universal approximation claim is not rigorously established.

- If GCFs truly simplify bilevel optimization, the paper should demonstrate clear advantages under nonlinear or non-Euclidean Φ functions (e.g., adversarial or transport-type cost functions).
However, all experiments use the trivial linear Φ(x, y) = ⟨x, y⟩, which reduces to the standard convex setting.
Hence, the results fail to support the “generalized” claim.

- The manuscript, in my ability, is difficult to read and overly terse. Many claims (e.g., “we show,” “we extend”) are stated without actual derivations or rigorous proofs, which makes the paper opaque to readers outside the narrow mathematical optimization community.

**Questions:**

see the Weaknesses

---

> ### Author Response · Authors · 2025-12-01
>
> Response to Reviewer 2 (MUph)
> Submission: 7643
>
> Dear Reviewer,
>
> Thank you very much for your detailed and critical review. Your comments on the theoretical assumptions, the role of generalized convexity, and the experimental design have been extremely helpful in improving the paper. I also sincerely apologize for the delayed response; as the sole author, implementing and validating the requested changes took substantial time.
>
> Below I respond to your points one by one.
>
> ------------------------------------------------------------
> Point 1
> ------------------------------------------------------------
> > In the introduction, the paper argues that GCFs can transform bilevel problems into single-level optimization problems, yet all subsequent examples (e.g., mechanism design and optimal transport) already presuppose the existence of GCF representations via the Φ-transform. This creates a circular argument: “We study GCFs because they simplify bilevel optimization; we know they simplify it because we assume the problem already admits a GCF form.”
>
> Response:
>
> You raise a valid concern. The intention was not to claim that GCF structure appears out of nowhere, but rather to:
>
> 1. Take problems where previous theoretical work had shown that they admit GCF representations, yet applied work has often treated them as bilevel problems; and
> 2. Show that once such a representation exists, our parameterization allows these problems to be solved numerically as single-level optimization problems rather than generic bilevel/min-max problems.
>
> In the revised manuscript, I have clarified this point in several ways:
>
> - In the Background and Applications sections, I explicitly treat the existence of GCF representations for optimal transport and mechanism design as standard results from the literature, with references to Singer, Rubinov, Villani, Ekeland, and others.
> - I emphasize that the role of GCFs here is two-fold:
>   (a) As a modeling lens already established in prior work for certain classes of problems; and
>   (b) As the object that we now parameterize and approximate in a way that enables practical learning and optimization.
>
> This separation should mitigate the impression of circularity: the existence of a GCF representation is taken from prior theory, and our contribution is to provide a finite, convex parameterization and universal approximation/differentiability results that make these representations usable in computational settings.
>
> ------------------------------------------------------------
> Point 2
> ------------------------------------------------------------
> > The notion of generalized convexity was formalized long ago in Singer (1997) and Rubinov (2013). The paper fails to clearly articulate whether its novelty lies in the parameterization of the function space or in an extension of existing approximation theorems. This ambiguity undermines the conceptual contribution.
>
> Response:
>
> I agree that the first version did not clearly separate the classical generalized convexity theory from the new contributions, even though older works such as Singer (1997) and Rubinov (2013) had been cited.
>
> In the revised paper:
> - I clarify that the novelty of this paper lies in:
>   - A finite, convex parameterization of the space of GCFs via finitely Y-convex functions, which generalizes max-affine parametrizations from the purely convex (inner-product) setting to general kernels Φ.
>   - Universal approximation theorems for both GCFs and their gradients, under mild assumptions (compact domains, local Lipschitzness, and equi-semiconvexity of the kernel).
>   - The application of these parameterizations to transform certain bilevel problems (OT, mechanism design) into single-level optimization tasks in practice.
>
> These points are now clearly stated in the Introduction and at the start of the Method section to address the ambiguity you identified.

---

> > ### Author Response · Authors · 2025-12-01
> >
> > ------------------------------------------------------------
> > Point 3
> > ------------------------------------------------------------
> > > Theorem 5.1 (GCF UAP) merely asserts that “Proposition 5.1 + 5.2 ⇒ density,” yet Proposition 5.1’s proof relies on Φ being globally Lipschitz. Since Φ is only assumed to be locally Lipschitz, there is no guarantee of a global constant on compact domains. → Therefore, the universal approximation claim is not rigorously established.
> >
> > Response:
> >
> > Thank you for pointing out this crucial issue. The whole paper works under the assumption of X and Y being compact, but I had forgotten to state this explicitly even though the proofs were using it.
> >
> > In the revised version:
> > - At the beginning of the Background section, I now explicitly assume that X and Y are compact subsets of Euclidean spaces.
> > - Under this compactness assumption, a locally Lipschitz kernel Φ is Lipschitz on X×Y, so the global Lipschitz constant used in the uniform approximation proof is well-defined.
> > - The reason I do this rather than assume global Lipschitzness directly is that many useful kernels are not globally Lipschitz on the unbounded domain but are locally Lipschitz.
> > - I updated the text to make this assumption explicit and consistent across the main text and the appendix proofs.
> >
> > With these changes, the universal approximation theorem in the generalized setting is now rigorously established under clear and standard assumptions (compact X,Y and locally Lipschitz Φ).
> >
> > ------------------------------------------------------------
> > Point 4
> > ------------------------------------------------------------
> > > If GCFs truly simplify bilevel optimization, the paper should demonstrate clear advantages under nonlinear or non-Euclidean Φ functions (e.g., adversarial or transport-type cost functions). However, all experiments use the trivial linear Φ(x, y) = ⟨x, y⟩, which reduces to the standard convex setting. Hence, the results fail to support the “generalized” claim.
> >
> > Response:
> >
> > I completely agree with this criticism of the original experiments. The first version relied only on a linear kernel, which did not showcase the benefits of generalized kernels.
> >
> > In response, I have substantially expanded the experimental section:
> >
> > 1. Added Optimal Transport experiments:
> >    - I now include experiments for both the classical quadratic cost ∥x−y∥² and the non-convex negative quadratic cost −∥x−y∥².
> >    - The negative quadratic cost lies outside the classical convex OT framework, but is naturally handled using generalized convex potentials.
> >    - The learned transport maps behave as expected in both cases, demonstrating that the generalized framework is genuinely useful beyond the linear/convex kernel.
> >
> > 2. Extended Mechanism Design experiments:
> >    - The auctions under the original linear kernel have now been scaled up to 250 items to show scalability.
> >    - Besides the linear kernel setting (Φ(x,y)=⟨x,y⟩), I introduce a more complex setting with a nonlinear kernel and nontrivial production costs and type distributions.
> >    - In this second setting, no analytic benchmark is known, but the learned mechanisms satisfy intuitive economic sanity checks (profit bounds, behavior as the number of goods increases).
> >
> > These additions directly address your concern by showing that the generalized kernel formulation is not only a theoretical generalization but also empirically useful in settings where Φ is non-linear or non-Euclidean.
> >
> > ------------------------------------------------------------
> > Point 5
> > ------------------------------------------------------------
> > > The manuscript, in my ability, is difficult to read and overly terse. Many claims (e.g., “we show,” “we extend”) are stated without actual derivations or rigorous proofs, which makes the paper opaque to readers outside the narrow mathematical optimization community.
> >
> > Response:
> >
> > I appreciate your candid assessment of the exposition. Unfortunately, the 10-page limit imposed by ICLR makes it challenging to include all such details. Nonetheless, in the revised version:
> > - Background and standard results (with references) are clearly separated from new contributions.
> > - Proofs are collected in the appendix with consistent labels, and the main text provides intuition, context, and references rather than detailed technical steps.
> > - Around each main theorem/proposition, I've added short informal explanations of:
> >   - Why the result is needed and what problem it addresses.
> >   - How it connects to other parts of the paper (e.g., density ⇒ gradient UAP ⇒ applications).
> > - In the Applications and Experiments sections, I rewrote the mechanism design part to make the bilevel nature of the original problem, the reduction via GCFs, and the learned quantities more explicit.
> >
> > While the paper remains mathematically oriented, I hope these revisions make it more accessible and less terse, particularly for readers outside the narrow core of generalized convex analysis.

---

> > > ### Author Response · Authors · 2025-12-01
> > >
> > > Once again, thank you for your detailed feedback and for helping strengthen both the theory and the empirical validation in the paper.

---

### Official Review · Reviewer_qwjF · 2025-10-30

**Soundness:** 2
**Presentation:** 2
**Contribution:** 2
**Rating:** 4
**Confidence:** 3

**Summary:**

This paper studies generalized convex functions (GCFs) and their gradients. The authors present the applications about the general GCFs, and then provide a new differentiable layer with a convex parameter space where it and its gradient are universal approximators for GCFs and their gradients
respectively. Finally, they also conduct experiments to demonstrate its effectiveness in learning optimal pricing mechanisms when selling multiple goods.

**Strengths:**

The overall framework of this paper is clear. The concept of generalized convexity could be an interesting topic for further exploration.

**Weaknesses:**

The theoretical contribution of this paper is limited. The work appears to summarize several properties related to generalized convexity but does not provide any particularly insightful perspectives. In addition, some important lemmas and theorems, such as Theorem 5.3, lack sufficient explanation. I believe that including several necessary remarks would be more helpful for readers.

**Questions:**

My main concern about this paper lies in its theoretical contribution. The current version appears to lack technical novelty. Specifically, Sections 3.2 and 5 seem to be a collection of properties related to generalized convexity. Could the authors provide a more detailed summary of the technical novelty and contributions?

Regarding the experimental section, the authors only conduct experiments on a few illustrative cases. I am wondering whether there are any experiments on real-world benchmarks, such as CIFAR or similar datasets.

---

> ### Author Response · Authors · 2025-12-01
>
> Response to Reviewer 1 (qwjF)
> Submission: 7643
>
> Dear Reviewer,
>
> Thank you very much for your thoughtful and constructive review. I believe your comments have helped significantly improve the paper. I also sincerely apologize for the delayed response; as the sole author, it took me some time to digest the comments and implement the necessary revisions.
>
> Below I respond to your points one by one.
>
> ------------------------------------------------------------
> Point 1
> ------------------------------------------------------------
> > The work appears to summarize several properties related to generalized convexity but does not provide any particularly insightful perspectives.
>
> Response:
>
> You are right that in the original version, the distinction between background material and new contributions was not sufficiently clear, which could make the generalized convexity part look like a mere summary.
>
> In the revised manuscript I have:
>
> - Explicitly marked the generalized convexity properties (such as the “Basic Properties” theorem and the generalized biconjugation theorem) as standard results from the literature (Singer, Rubinov, etc.), and moved properties of generalized convexity and their full proofs to the appendix for completeness rather than presenting them as part of the main text.
> - Clearly separated the expository “Background” and “Applications” sections from the “Method: Parameterizing Generalized Convex Functions” section, where the new results are concentrated.
> - Emphasized in the Introduction and in Section 5 that the main novelty lies in:
>   (i) A convex, finite-dimensional parameterization of generalized convex functions via finitely Y-convex functions; and
>   (ii) Universal approximation results for both generalized convex functions and their gradients under mild semiconvexity assumptions on the kernel.
>
> I believe what remains from Sections 3 and 4 (Background and Applications) is the minimum necessary for understanding the theory and its applications.
>
> ------------------------------------------------------------
> Point 2
> ------------------------------------------------------------
> > Some important lemmas and theorems, such as Theorem 5.3, lack sufficient explanation. I believe that including several necessary remarks would be more helpful for readers.
>
> Response:
>
> I agree that the original exposition around some key lemmas and theorems, especially in Section 5, was terse.
>
> In the revised version:
>
> - I expanded the explanations immediately before and after the main propositions and theorems in the Method section.
>   - For the semiconvexity definition, I added an intuitive explanation of why it is weaker than convexity and why it is the right condition for gradient approximation (it rules out “downward kinks” while allowing finite negative curvature).
>   - For the gradient stability proposition, I explicitly explain that it is the bridge from uniform function approximation to uniform gradient approximation under a common semiconvexity constant.
>   - For the gradient universal approximation theorem, I clearly spell out the logic: density of finitely Y-convex functions + equi-semiconvexity of the kernel + gradient stability ⇒ density of gradients.
> - In the appendix, I organized and labeled the proofs of these results more systematically, with clear cross-references from the main text.
>
> These additions are aimed at making the theoretical contributions more transparent and easier to follow for readers who are not specialists in generalized convex analysis.
>
> ------------------------------------------------------------
> Point 3
> ------------------------------------------------------------
> > Sections 3.2 and 5 seem to be a collection of properties related to generalized convexity.
>
> Response:
>
> This is closely related to your first comment. In the revision, I have restructured and clarified those sections to avoid the impression that they are just collections of properties without clear purpose.
>
> Concretely:
>
> - Section 3 (“Background”) is now explicitly framed as expository, summarizing the classical generalized convexity framework and key identities needed later.
> - The new contributions are collected in Section 5 (“Method: Parameterizing Generalized Convex Functions”). There I emphasize that:
>   - The density of finitely Y-convex functions in the space of Y-convex functions is new in this generalized setting when combined with the conditions we impose on the kernel.
>   - The universal approximation theorem for gradients under semiconvexity is the main technical contribution on the gradient side.
>
> I also provide explicit references for the background properties, which further separates them from the new results.

---

> > ### Author Response · Authors · 2025-12-01
> >
> > ------------------------------------------------------------
> > Point 4
> > ------------------------------------------------------------
> > > Could the authors provide a more detailed summary of the technical novelty and contributions?
> >
> > Response:
> >
> > Yes. In the revised Introduction, I have rewritten the “Contributions” paragraph to be more explicit and precise. The main technical contributions are now summarized as:
> >
> > 1. A finite, differentiable parameterization of generalized convex functions (GCFs) with a convex parameter space, via finitely Y-convex functions, enabling stable first-order optimization.
> > 2. New universal approximation results for both GCFs and their gradients under mild regularity assumptions (local Lipschitzness and equi-semiconvexity) on the surplus kernel Φ. In particular, I want to emphasize that approximation results on gradients are much harder to obtain, making this result particularly useful.
> > 3. A neural network interpretation that connects finitely Y-convex parameterizations to shallow architectures with max aggregation, suggesting deeper analogues.
> > 4. Practical instantiations in optimal transport and multi-item auction design, including non-convex and nonlinear kernels where classical convex methods do not directly apply.
> >
> > These points are now stated clearly up front, and the Method and Experiments sections are organized to correspond to this list.
> >
> > ------------------------------------------------------------
> > Point 5
> > ------------------------------------------------------------
> > > I am wondering whether there are any experiments on real-world benchmarks, such as CIFAR or similar datasets.
> >
> > Response:
> >
> > I appreciate this request for broader empirical evaluation. Unfortunately, to my knowledge there are no standard benchmarks that directly match our setting, which is more abstract and not simple classification. This is why the original application focused on mechanism design, where we had some intuition about what the optimal mechanism should look like.
> >
> > In the revised paper, I have significantly expanded the experiments:
> >
> > 1. Added Optimal Transport experiments:
> >    - I now include experiments where the cost is both the classical quadratic cost ∥x−y∥² and a non-convex “negative quadratic” cost −∥x−y∥².
> >    - The latter is particularly challenging and lies outside the reach of standard convex OT solvers, but is naturally handled in the generalized convexity framework.
> >    - The learned transport maps match the intuitive behavior under both costs (short moves for quadratic cost, long moves for negative quadratic), and the marginals are well matched.
> >
> > 2. Extended Mechanism Design experiments:
> >    - For the original linear setting I have extended the dimensionality of the problem to 250 items, showing the scalability of the method.
> >    - I have added a second, more complex setting with nonlinear valuation kernels and production costs. As mentioned before, we do not have theoretical results on how the auction should look in this setting. To compensate, we note that the learned auction exhibits the sanity checks one would expect.
> >
> > These domains were chosen because GCF structure is central and theoretically well-understood. Applying the method directly to CIFAR-style image benchmarks would require designing structured kernels Φ and architectures suited to high-dimensional images (e.g., convolutional or equivariant kernels), which I view as an important direction for future work rather than something I could robustly add in this revision.
> >
> > Once again, thank you for your insightful feedback and for helping improve the clarity and scope of the paper.

---

### Meta-Review · Area_Chair_iDkc · 2025-12-14

**Summary:**

The reviewers raised the following concerns:
- Limited novelty as compared to existing studies
- Challenges in extending the scalar product by the function $\Phi$
- Lack of experiments on real-world benchmarks, such as CIFAR or similar datasets
- The universal approximation claim is not rigorously established
- A circular argument on GCFs and bilevel optimization problems
- All experiments use the trivial linear $\Phi$ functions
- Several statements are stated without actual derivations or rigorous proofs
- Narrow empirical validation as the paper does not showcase improvements in adversarial training, optimal transport, or robust optimization

**Reviewer Concerns:**

According to the authors' response, I think the following concerns are well addressed
- Narrow empirical validation (the author has added experimental results on optimal transport)
- The universal approximation claim is not rigorously established (the author has mentioned that the paper considers compact domains)
- A circular argument on GCFs and bilevel optimization problems (the author has added some explanations)
- Several statements are stated without actual derivations or rigorous proofs (the author has made changes to clarify the statements)

The following concerns are not well addressed:
- Lack of experiments on real-world benchmarks, such as CIFAR or similar datasets (the author did not add experimental results on real-world benchmarks)
- All experiments use the trivial linear $\Phi$ functions (while the author has added nonlinear $\Phi$ functions, the dimension is very small in the experiments (no larger than 20). Furthermore, the kernel is additive in different cardinalities and therefore cannot capture the interaction among different features)
- Challenges in extending the scalar product by the function $\Phi$ (the author did not clarify the challenges in the response)

**Reviewer Scores:**

Reviewer PtpT may not change his/her score since the concern in the challenges in considering $\Phi$ is not well addressed.

Reviewer MUph may not change his/her score since the concern of using linear kernels in all the experiments is not well addressed

Reviewer qwjF may not change his/her score since the author did not add experimental results in real-world benchmarks

---

### Decision · Program_Chairs · 2026-01-26

Reject